# Prevalence and risk factors of coliform-associated mastitis and antibiotic resistance of coliforms from lactating dairy cows in North West Cameroon

Ursula Anneh Abegewi[1], Seraphine Nkie Esemu[1,2], Roland N. Ndip[1,2], Lucy M. Ndip[1,2]*

1 Department of Microbiology and Parasitology, Faculty of Science, University of Buea, Buea, Cameroon,
2 Laboratory for Emerging Infectious Diseases, University of Buea, Buea, Cameroon

* lndip@yahoo.com

## Abstract

### Background

Coliform bacteria are major causative agents of bovine mastitis, a disease that has devastating effect on dairy animal health and milk production. This cross-sectional study, carried out in the North West region of Cameroon, sought to determine the prevalence of bovine mastitis, coliforms associated with bovine mastitis, risk factors for infection and the antibiotic resistance pattern of coliform bacterial isolates.

### Materials and methods

A total of 1608 udder quarters were sampled from 411 cows using a questionnaire, clinical examination, California Mastitis Test and milk culture. Primary isolation of coliform bacteria was done on MacConkey agar while identification of coliforms employed Gram-staining and biochemical testing. Each coliform bacterial isolate was challenged with 11 antibiotics using the Kirby-Bauer disc diffusion method.

### Results

The prevalence of mastitis was 53.0% (218/411) and 33.1% (532/1608) at the cow- and quarter-levels respectively. Overall, 21.9% (90/411) cows and 8.2% (132/1608) udder quarters showed coliform mastitis. *Escherichia coli* was isolated in 7.0% of mastitis milk, and other coliforms isolated were *Enterobacter cloacae* (12.6%), *Klebsiella pneumoniae* (2.4%), *Enterobacter sakazakii* (1.1%), *Klebsiella oxytoca* (0.8%), *Citrobacter freudii* (0.4%), *Serratia ficaria* (0.4%) and *Serratia liquefaciens* (0.2%). Lactation stage, breed, history of mastitis and moist/muddy faeces contaminated environment were significantly associated (*P*-value < 0.05) with coliform mastitis. Coliform isolates (99.0%; 203/205) were resistant to at least one antibiotic tested. Amoxicillin had the highest resistance (88.8%) while norfloxacin had the least resistance (3.4%). Multidrug resistance was exhibited by 52.7% (108/205) of the isolates in a proportion of 27.8% *Enterobacter cloacae*, 10.7% *E. coli*, 6.3% *Klebsiella*

**Data Availability Statement:** All relevant data are within the paper.

**Funding:** The author(s) received no specific funding for this work.

**Competing interests:** The authors have declared that no competing interests exist.

*pneumoniae*, 2.9% *Enterobacter sakazakii*, 2.0% *Klebsiella oxytoca*, 1.0% *Citrobacter freundii*, 1.0% *Serratia ficaria*, 0.5% *Serratia liquefaciens* and 0.5% *Serratia odorifera*.

## Conclusion

Results indicate a need to educate these dairy farmers about mastitis (particularly subclinical), proper hygiene methods in milking and the public health implications of consuming contaminated raw milk.

## 1. Introduction

Bovine mastitis, an inflammation of the cow's mammary gland, is the most common and most costly dairy cow disease worldwide [1, 2]. It adversely affects the business of milk production with substantial financial losses [3, 4] stemming from decreased quantity and quality of milk produced [5].

The majority of mastitis cases are of bacterial origin, with coliform bacteria being among the few predominant causative pathogens [6, 7]. Coliforms frequently implicated are *Escherichia coli (E. coli)*, *Enterobacter* species, *Klebsiella* species, *Serratia* species and *Citrobacter* species [8, 9] with *E. coli* responsible for more than 80.0% of coliform mastitis cases [7, 10].

Coliform bacteria inhabit the gastrointestinal tract of most animals. These microorganisms are ubiquitous in the cow's environment and cannot be easily eradicated even in well-managed dairy herds [11]. They mostly infect the mammary gland primarily through the exposure of the teat end to moisture, mud and faecal material, and exposure may repeatedly occur [6]. Coliform bacteria do not have a predilection for the mammary gland, thus are opportunistic pathogens. After infection with coliform bacteria, manifestations can be clinical, subclinical, or the bacteria may remain dormant. The severity of clinical signs ranges from a mild or moderate type of disease predominantly localized around the udder, to a more severe systemic illness, due to bacteraemia and septicaemia which may be fatal [8, 12]. About 60.0–70.0% of systemic clinical signs are associated with coliform infections [8]. Chronic coliform infections also occur and may be subclinical but typically elicit recurrent clinical episodes [13].

Most coliform infections are self-limiting, but if not cleared by the cow's immune system, they may lead to unlimited growth of the coliform bacteria in the udder, making the use of antibiotic therapy, one of the essential methods to control the infection from advancing to complications [14, 15]. However, the report of antibiotic resistance of mastitis coliform bacterial isolates is a cause for concern [10, 16–18]. The frequent treatment failure, increased severity of the disease, and the ability of the bacteria to spread and transfer resistant phenotype to human populations via unpasteurized milk is a growing concern [19]. Monitoring antibiotic resistance in coliform bacteria is, therefore, important for public health reasons.

In Cameroon, cattle constitute the primary source of fresh milk, with 17.9% of households operating on dairy farming [20]. Similar to other sub-Saharan countries, milk production in Cameroon is dominated mostly by small-scale farmers [21] who get their income and nutrition from milk and milk products. Thus, dairying contributes significantly to alleviating poverty and reducing malnutrition, particularly in rural and peri-urban areas and serves as a source of income for the small-scale farmers who are primarily women [22]. Therefore, coliform mastitis can have huge economic impact on these small-scale famers who may not withstand the enormous financial losses associated with the disease. Moreover, apart from economic consequences associated with mastitis, there are also public health implications that

have been linked to the handling and consumption of unpasteurized coliform-contaminated milk and/or milk products by humans [8, 23]. Milk could be contaminated by the Shiga toxigenic *E. coli* strain, which is risky due to their great zoonotic importance [24].

So far, there is limited information about coliform bacteria in bovine mastitis cases in the North West Region of Cameroon. To the best of our knowledge, the prevalence, risk factors and antibiotic resistance profile of coliform bacteria associated with bovine mastitis cases have not been investigated in our study area. Therefore, this study sought to determine the overall prevalence of bovine mastitis, isolate, identify and establish the prevalence of coliforms associated with bovine mastitis, particularly *E. coli*, elucidate risk factors for infection and determine antibiotic resistance pattern of coliform isolates from cow milk. The results of this study will provide important epidemiological data on bovine coliform-associated mastitis.

## 2. Materials and methods

### 2.1 Ethical considerations

The use of animals in the study was approved by the North West Regional Delegation of Livestock, Fisheries and Animal Industries. The objectives of this study were explained orally to dairy farmers, and their consent was obtained to participate in the study. Questionnaire responses about the dairy cows were given voluntarily, and farmers were permitted to withdraw their consent at any time.

### 2.2 Study area

The study was carried out in all seven administrative divisions in the North West region of Cameroon (Fig 1). These administrative divisions are: Boyo, Bui, Donga-Mantung, Menchum, Mezam, Momo and Ngoketunjia. The North West region is situated in the western highlands of Cameroon. It occupies a surface area of about 17, 300km$^2$ between latitudes 5˚ 20' and 7˚ 15' N and longitudes 9˚ 30' and 11˚ 15' E at an altitude of 300 to 3000m above sea level. The region is characterized by two distinct seasons: a dry season from mid-October to mid-March and a rainy season from mid-March to mid-October, an average rainfall of about 2400mm and a daily average temperature of 23˚C with a range between 15˚C and 32˚C [25]. Its agro-climatic conditions favour cattle rearing, thereby making the region one of the most important cattle production areas in Cameroon and one of Cameroon's most favourable milk production areas [26].

### 2.3 Study design and population

A cross-sectional study was carried out to address the objectives of this study. Cattle herds were selected randomly based on accessibility to the farm and farmers' willingness to participate. Data were collected from all lactating cows in each selected herd except those cows that had received antibiotics within the past 15 days.

The study included lactating cows used for dairy purposes (including cows raised for both milk and beef). The cows were reared under three husbandry systems; the extensive system where cows are left to wander and graze during the day but are enclosed at night, the semi-intensive system where cows graze freely on pasture but receive supplementary feeds, particularly during milking and are enclosed at night and the intensive system where cows remain confined and are catered for.

The cow breeds included in the study were: local African breeds consisting of Gudali, red Fulani, white Fulani and Boran breeds; pure exotic breeds consisting only Holstein-Friesian;

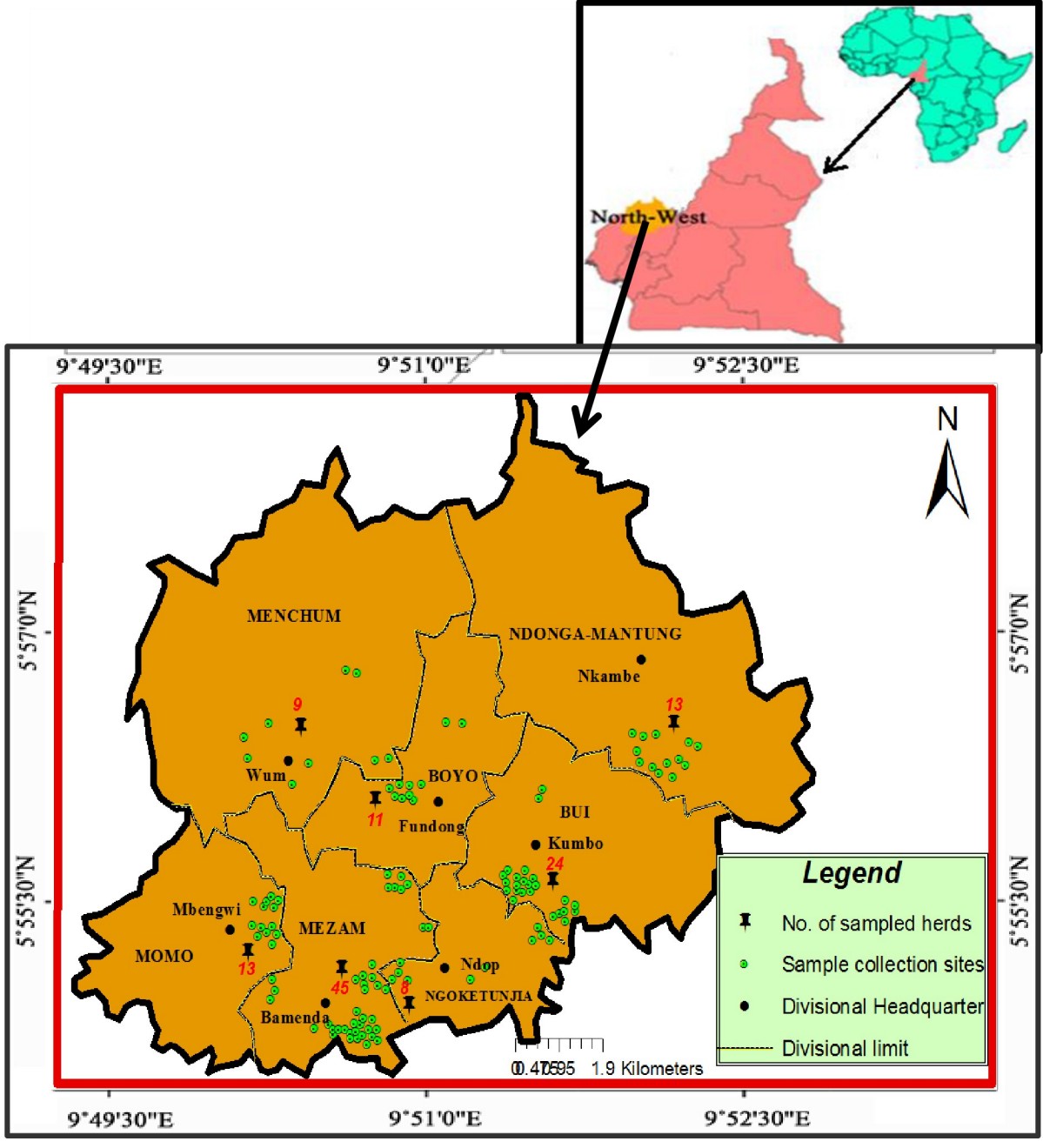

**Fig 1. Map of the study area showing sample collection sites.**

and cross-breeds consisting of exotic (Holstein, Jersey, Simmental, Charolais and Brahman) and local species.

## 2.4 Determination of sample size

In the absence of documented evidence of a similar study in this area, the sample size was calculated from the formula (n = $Z^2 * P(1-P)/d^2$) recommended by Thrusfield [27], with a 95% confidence interval, at 5% desired absolute precision and expected prevalence of 50%. Where

$n$ = sample size, $Z$ = α value of 95% confidence interval = 1.96, $P$ = expected prevalence in population-based on previous studies, $d$ = desired absolute error or precision = 5%. Hence, the expected minimum number of lactating dairy cows to be included in the study was 384.

## 2.5 Data collection

Data were collected from March to November 2019. Basic information on cow and herd management practices was collected using a semi-structured pretested questionnaire. Specifically, data captured in the questionnaire included cow information (such as age, breed, parity, stage of lactation and previous history of mastitis), herd size, husbandry system, method of milking, number of times cow was milked per day, cleanliness of cow environment and floor type. A trained veterinarian examined each cow for signs of clinical mastitis, which included evidence of pain, swelling, and milk changes such as the presence of clots, change of colour and consistency of milk, fever and body weakness [9]. The blindness of teats was also recorded.

## 2.6 California mastitis test

The California Mastitis Test (CMT) (ImmuCell® CMT, Portland) was performed for each quarter milk sample as a screening test for inflammation, particularly subclinical mastitis, as previously described [28, 29]. Briefly, two millilitres of quarter fore-milk and an equal amount of CMT reagent were collected into corresponding shallow wells of the CMT plastic paddle. A gentle circular motion was applied to the mixture in a horizontal plane for 15 seconds, and the result was scored as N (negative), T (trace), 1 (weakly positive), 2 (distinct positive) and 3 (strongly positive) based on thickening or gel formation. A quarter was positive if it had a score of $\geq 1$, and a cow was considered positive when at least one quarter had a positive CMT score.

## 2.7 Milk sample collection for bacteriology

Quarter milk was collected aseptically from the teat into a sterile tube according to the procedure described by National Mastitis Council (NMC) [30]. Milk collection was done early in the morning before milking. The veterinarian's hands were washed, disinfected and a new pair of gloves was worn before milk collection from each cow. The cow's udder, especially teat, was thoroughly washed with clean running water and dried with disposable paper towels. Later, the teats were disinfected with cotton wool soaked in 70% ethanol and allowed to dry. A ball of separate cotton wool was used for each teat. The first few streams of milk were discarded, and approximately 10mL of milk were directly stripped from teats into pre-labelled screw-capped sterile plastic tubes. The samples were transported at +4°C in a cool box and stored at -20°C in the Laboratory for Emerging Infectious Diseases, University of Buea, for analysis.

**2.7.1 Isolation and identification of coliforms.** Each quarter milk sample was cultured for bacterial isolation using standard procedures [30]. An aliquot of 10μl of milk sample was inoculated by streaking on MacConkey agar (Liofilchem Diagnostic, Italy). Each frozen milk sample was thawed only once at room temperature and mixed vigorously. Unless otherwise stated, all incubations were done at 37°C for 24–48h. Colonial morphology, lactose fermentation and Gram reaction aided identification. Gram-negative bacteria were subcultured on eosin methylene blue for presumptive identification of *E. coli*. Presumptive colonies were subcultured on nutrient agar to get pure cultures. All Gram-negative bacterial isolates were subjected to the Analytical Profile index for Enterobacteriaceae (API 20E) (Biomerieux, 2002, France) testing following manufacturer's instructions supported with oxidase testing using oxidase strips (Sigma, Switzerland) for confirmatory identification of coliforms.

**2.7.2 Antibiotic susceptibility testing by disc diffusion technique.** *In vitro* antibiotic susceptibility testing was done by Kirby-Bauer disc diffusion method on Mueller-Hinton agar (Liofilchem Diagnostic Srl, Italy) according to the Clinical and Laboratory Standards Institute [31]. Each of the coliform bacterial isolates was challenged with commonly used antibiotics in animal and human health. A panel of 11 antibiotics (Oxoid, England) was used and included: beta-lactams [ampicillin (10µg), amoxicillin (10µg), cephalothin (30µg)]; aminoglycosides [gentamicin (10µg), streptomycin (10µg)]; tetracyclines [tetracycline (30µg)]; quinolones [norfloxacin (5µg), nalidixic acid (30µg)]; folate inhibitors [cotrimoxazole (25µg)]; phenicols [chloramphenicol (10µg)] and macrolides [erythromycin (30µg)]. The turbidity of each bacterial inoculum was adjusted to that of 0.5 McFarland standard and then inoculated on Mueller-Hinton agar. The antibiotic discs were applied gently on the agar using sterile forceps, and plates were incubated at 35˚C for 18h. Growth inhibition diameter was measured in millimetres and the data were interpreted using Clinical and Laboratory Standards Institute criteria [31] to classify isolates as susceptible, intermediate or resistant (Table 1). Isolates that were resistant to at least one antibiotic in three or more classes were termed multidrug-resistant [32].

## 2.8 Data analysis

The data generated from the field, laboratory analyses and questionnaire survey were entered into Microsoft Excel spreadsheets 2010 and was analyzed using MINITAB version 17 statistical package. Prevalence was calculated as a percentage value of the proportion of positive cases against total number sampled. The association between the explanatory/dependent and response/independent variables was analyzed using the chi-square test. Multivariate logistic regression analysis was employed to analyze the relative strength of the different risk factors on coliform-associated mastitis. The independent variables included in the model were those that showed statistical significance ($P < 0.05$) in the initial chi-square test. The model was assessed for goodness-of-fit using the Hosmer-Lemeshow test. Statistical differences were considered significant when $P < 0.05$.

## 3. Results

### 3.1 Characteristics of herds and cow population sampled

A total of 123 herds with a mean herd size of 13 (range: 1–102 cattle) were sampled, comprising 47.2% (58/123) that practised intensive farming, 5.7% (7/123) that practiced semi-intensive

Table 1. Antibiotic susceptibility test interpretative criteria and cut-off values for coliforms [31].

| Antibiotic | Symbol | Disc potency | Interpretative categories and zone diameter breakpoints | | |
|---|---|---|---|---|---|
| | | | Susceptible | Intermediate | Resistant |
| Amoxicillin | AMX | 10µg | ≥21 | 14–20 | ≤13 |
| Ampicillin | AMP | 10µg | ≥17 | 14–16 | ≤13 |
| Cephalothin | KF | 30µg | ≥18 | 15–17 | ≤14 |
| Gentamicin | CN | 10µg | ≥15 | 13–14 | ≤12 |
| Streptomycin | STR | 10µg | ≥15 | 12–14 | ≤11 |
| Tetracycline | TET | 30µg | ≥19 | 15–18 | ≤14 |
| Nalidixic acid | NAL | 30µg | ≥19 | 14–18 | ≤13 |
| Norfloxacin | NOR | 5µg | ≥17 | 13–16 | ≤12 |
| Cotrimoxazole | COT | 1.25/23.75 µg, (25µg) | ≥16 | 11–15 | ≤10 |
| Chloramphenicol | CHL | 10µg | ≥18 | 13–17 | ≤12 |
| Erythromycin | ERY | 30µg | ≥23 | 14–22 | ≤13 |

farming and 47.2% (58/123) that practised extensive farming. Milking was done manually in all herds, twice a day in 65 (52.8%) and once a day in the remaining 58 (47.2%). In 52.8% (65/123) of the herds (which included cows raised under the intensive and semi-intensive farming system), the udder was washed with clean running water before milking while 47.2% (58/123) of the herds (which included cows raised under the extensive farming system) did not clean the udder before milking. Gloves were not worn before milking in any of the herds and the teats were neither dipped into any disinfectant nor cleaned after milking. Four hundred and eleven lactating cows with a mean age of 6.4 years (age range 2–15 years), mean lactation number of 3 calves (range 1–11 calves) and mean lactation length of 5.4 months (range 1–12 months) were sampled, and a majority (22.4%) was local breeds

### 3.2 Clinical examination and CMT results

Upon examining the quarters of the 411 cows, it was observed that 36 out of 1644 quarters (2.2%) were nonfunctional (blind), and 5.8% of the 1608 functional quarters (within 33 cows) displayed clinical signs characterized by watery milk or clots, flakes or blood and swelling or sore udder/teat, with only two of these cows presenting additional signs of fever, depression and lack of appetite.

The remaining 1512 functional quarters without clinical signs had varied CMT scores (Fig 2). All the 96 quarters with evidence of clinical mastitis were positive by CMT.

### 3.3 Prevalence of mastitis and coliform-associated mastitis

Table 2 depicts the prevalence of mastitis and coliform-associated mastitis in North West Cameroon. Out of the 411 cows examined, 53.0% (218/411) cows and 33.1% (532/1608) quarters had evidence of mastitis. The prevalence of subclinical mastitis was significantly higher compared to clinical mastitis ($P = 0.000$). The prevalence of coliform-associated mastitis at cow-level was 21.9% (90/411) and 8.2% (132/1608) at quarter-level. The proportion of coliform

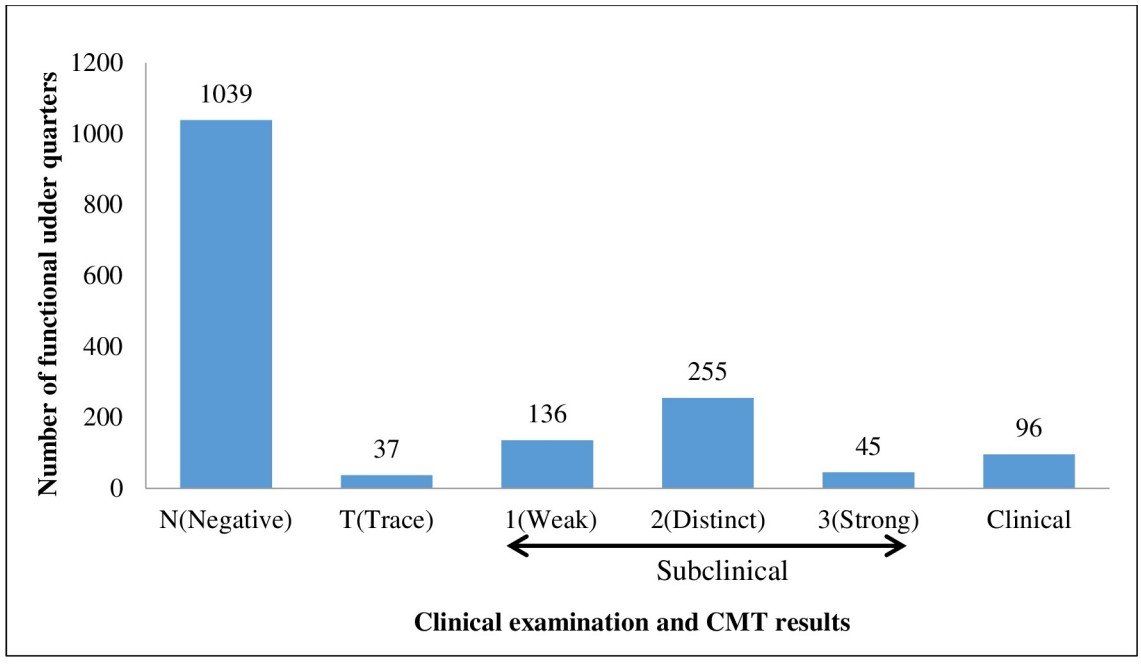

**Fig 2. Clinical examination and CMT results for functional quarters.**

**Table 2. Prevalence of mastitis and coliform-associated mastitis in North West Cameroon.**

| Level | Number examined | Number positive for mastitis (%) | | Total (%) | Number positive for coliform-associated mastitis (%) | | Total (%) |
|---|---|---|---|---|---|---|---|
| | | Clinical | Subclinical | | Clinical | Subclinical | |
| Cow | 411 | 33 (8.0)* | 185 (45.0)* | 218 (53.0)[a] | 15 (3.7) | 75 (18.2) | 90 (21.9)[b] |
| Quarter | 1608 | 96 (6.0)# | 436 (27.1)# | 532 (33.1)[c] | 18 (1.1) | 114 (7.1) | 132 (8.2)[d] |

[a] 95% confidence interval: 53.0 ± 4.8%

[b] 95% confidence interval: 33.1 ± 2.3%

[c] 95% confidence interval: 21.9 ± 4.0%

[d] 95% confidence interval: 8.2 ± 1.3%

*($X^2$ = 144.233, P = 0.000)

#($X^2$ = 260.363, P = 0.000)

bacteria associated with mastitis at cow-level was higher (45.45%, 15/33) in clinical mastitis cases compared to subclinical cases (40.5%; 75/185). Conversely, the proportion at the quarter-level was higher (26.0%; 114/436) in subclinical cases than in clinical cases (18.8%; 18/96). However, the difference at both levels was not statistically significant. Among the herds visited, 78.9% (97/123) had at least a cow positive for mastitis.

## 3.4 Occurrence of coliform bacteria in milk

The occurrence of coliform bacteria in all 1608 quarter milk samples was 12.8% (205/1608), and in mastitis samples, the occurrence was 24.8% (132/532). Even though coliform bacteria were isolated from non-mastitis milk samples, their isolation from mastitis samples was significantly (P = 0.000) higher at cow-level (41.3%; 90/218) and quarter-level (24.6%; 132/532) compared with non-mastitis samples (Table 3). The most frequently isolated coliform bacteria in mastitis quarters were *Enterobacter cloacae* (12.6%) and *Escherichia coli* (7.0%) (Table 4).

Among the 90 coliform-associated mastitis cows, a majority (62.2%, 56/90) had only one of its quarters infected, while only 1.1% (1/90) cow had all four quarters infected (Fig 3). Among the 27 coliform-associated mastitis cows with two quarters infected, 23 of the cows had all two quarters infected with the same coliform bacteria while the remaining 4 cows had different coliform bacteria. For the 6 coliform-associated mastitis cows that had three quarters infected, 4 cows had all three quarters infected with the same coliform bacteria while the remaining 2 had two different coliforms. In the one coliform-associated mastitis cow, the same coliform bacteria infected all four of its quarters.

## 3.5 Coliform-associated mastitis risk factors

Among the risk factors investigated in this study, chi-square analysis revealed that the cow-level prevalence of coliform mastitis was significantly (P < 0.05) associated with lactation stage, breed, history of mastitis and moist/muddy faeces contaminated environment, as shown

**Table 3. Coliform bacterial isolation in mastitis and non-mastitis cows.**

| Clinical state | Cow-level | | | Quarter-level | | |
|---|---|---|---|---|---|---|
| | Number examined | Number of coliform positive (%) | *P*-value | Number examined | Number of coliform positive (%) | *P*-value |
| Mastitis | 218 | 90 (41.3) | 0.000 | 532 | 132 (24.8) | 0.000 |
| No mastitis | 193 | 38 (19.7) | | 1076 | 73 (6.8) | |
| Total | 411 | 128 (31.1) | | 1608 | 205 (12.8) | |

**Table 4. Occurrence of coliform bacterial isolates in quarter milk samples.**

| Mastitis state | Number of milk samples cultured | Number of coliform bacterial—culture positive quarter milk samples (%) | | | | | | | | | Total number of coliform positive quarters (%) |
|---|---|---|---|---|---|---|---|---|---|---|---|
| | | *Citrobacter freundii* | *Enterobacter cloacae* | *Enterobacter sakazakii* | *Escherichia coli* | *Klebsiella oxytoca* | *Klebsiella pneumoniae* | *Serratia ficaria* | *Serratia liquefaciens* | *Serratia odorifera* | |
| Clinical | 96 | 0 (0.0) | 7 (7.3) | 1 (1.0) | 8 (8.3) | 1 (1.0) | 1 (1.0) | 0 (0.0) | 0 (0.0) | 0 (0.0) | 18 (18.8) |
| Subclinical | 436 | 2 (0.5) | 60 (13.8) | 5 (1.2) | 29 (6.7) | 3 (0.7) | 12 (2.8) | 2 (0.5) | 1 (0.2) | 0 (0.0) | 114 (26.1) |
| **Total mastitis** | **532** | **2 (0.4)** | **67 (12.6)** | **6 (1.1)** | **37 (7.0)** | **4 (0.8)** | **13 (2.4)** | **2 (0.4)** | **1 (0.2)** | **0 (0.0)** | 132 (24.8) |
| No mastitis | 1076 | 0 (0.0) | 60 (5.6) | 4 (0.4) | 0 (0.0) | 6 (0.5) | 2 (0.2) | 0 (0.0) | 0 (0.0) | 1 (0.1) | 73 (6.8) |
| **Total** | **1608** | **2 (0.1)** | **127 (7.9)** | **10 (0.6)** | **37 (2.3)** | **10 (0.6)** | **15 (0.9)** | **2 (0.1)** | **1 (0.1)** | **1 (0.1)** | 205 (12.8) |

in Table 5. Multivariate logistic regression analysis of risk factors significantly associated with coliform mastitis prevalence revealed that all the variables entered remained significant predictors of coliform mastitis ($P < 0.05$) (Table 6). The Hosmer-Lemeshow Goodness-of-Fit test suggested that the model fitted the data ($X^2 = 9.44$, $P = 0.150$).

## 3.6 Antibiotic resistance of coliform bacterial isolates

Out of the 205 coliform isolates, 203 (99.0%) exhibited resistance to at least one antibiotic tested, with 10 (4.9%) of the isolates exhibiting resistance to eight antibiotics. The highest resistance was against amoxicillin (88.8%), followed by cephalothin (75.1%) and erythromycin (61.0%), whereas the least resistance was observed with norfloxacin (3.4%), followed by gentamicin (10.2%), nalidixic acid (13.8%) and cotrimoxazole (19.0%) (Table 7).

Among the coliform isolates tested, 15.1% (31/205) and 31.2% (64/205) were resistant to one and two antibiotic classes, respectively. Resistance to three or more classes, multidrug resistance (MDR), was exhibited by 52.7% (108/205) isolates (in the proportion of: 27.8% (57/

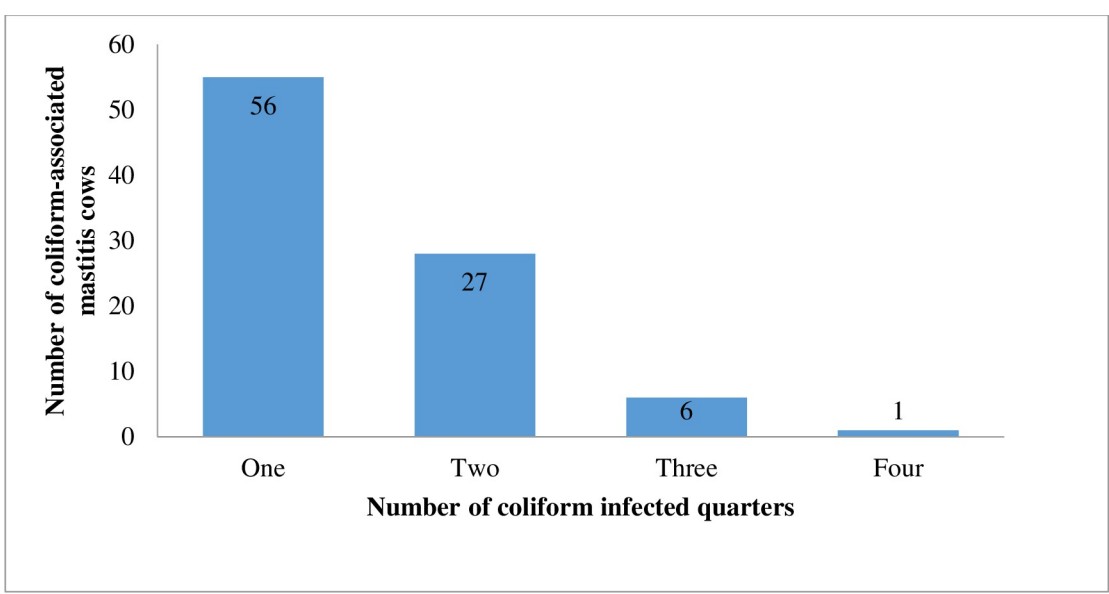

**Fig 3. Number of quarters infected for coliform-associated mastitis cows.**

**Table 5. Risk factors associated with the prevalence of coliform mastitis.**

| Risk factor | Category | Number of cows examined | Number of positive cows | Coliform mastitis Prevalence (%) | Chi square | P-value |
|---|---|---|---|---|---|---|
| Age (years) | 2–5 | 161 | 31 | 19.25 | 1.639 | 0.441 |
| | > 5–9 | 203 | 46 | 22.66 | | |
| | > 9 | 47 | 13 | 27.66 | | |
| Parity | 1 | 73 | 10 | 13.70 | 5.620 | 0.060 |
| | 2 | 124 | 24 | 19.35 | | |
| | ≥ 3 | 214 | 56 | 26.17 | | |
| Lactation stage | Early (≤ 2 months) | 112 | 39 | 34.82 | | |
| | Mid (3–6 months) | 110 | 13 | 11.82 | 17.827 | **0.000*** |
| | Late (> 6 months) | 189 | 38 | 20.11 | | |
| Breed | Local | 250 | 41 | 16.40 | 11.872 | **0.003*** |
| | Pure exotic (Holstein) | 92 | 26 | 28.26 | | |
| | Exotic and local cross | 69 | 23 | 33.33 | | |
| History of mastitis | No | 356 | 70 | 19.66 | 7.769 | **0.005*** |
| | Yes | 55 | 20 | 35.36 | | |
| Husbandry system | Intensive | 79 | 19 | 24.05 | 0.677 | 0.713 |
| | Semi-intensive | 35 | 6 | 17.14 | | |
| | Extensive | 297 | 65 | 21.89 | | |
| Moist/muddy faeces contaminated environment | No | 62 | 5 | 8.06 | 8.169 | **0.004*** |
| | Yes | 349 | 85 | 24.36 | | |
| Floor type | Concrete | 45 | 11 | 24.44 | 0.192 | 0.662 |
| | Earth | 366 | 79 | 21.58 | | |
| Herd size | <5 cattle | 74 | 19 | 25.7 | 0.809 | 0.667 |
| | 5–10 cattle | 11 | 2 | 18.2 | | |
| | >10 cattle | 326 | 69 | 21.2 | | |

*Statistically significant variables ($P < 0.05$)

**Table 6. Multivariate analysis of risk factors associated with coliform mastitis.**

| Risk factor | Category | Number of positive cows (%) | COR (95% CI) | AOR (95% CI) | P-value |
|---|---|---|---|---|---|
| Lactation stage | Early (≤ 2 months) | 39 (34.82) | 3.9863 (1.9849, 8.0057) | 4.3289 (2.1004, 8.9220) | **0.000*** |
| | Mid (3–6 months) | 13 (11.82) | 1 | 1 | |
| | Late (> 6 months) | 38 (20.11) | 1.8903 (0.9581, 3.7295) | 2.3747 (1.1787, 4.7845) | |
| Breed | Local | 41 (16.40) | 1 | 1 | **0.022*** |
| | Holstein–Friesian | 26 (28.26) | 2.0081 (1.1426, 3.5295) | 1.2766 (0.6748, 2.4152) | |
| | Exotic–local cross | 23 (33.33) | 2.5488 (1.3958, 4.6543) | 2.5050 (1.3190, 4.7573) | |
| History of mastitis | No | 70 (19.66) | 1 | 1 | **0.044*** |
| | Yes | 20 (35.36) | 2.3347 (1.2706, 4.2899) | 2.0493 (1.0289, 4.0819) | |
| Moist / muddy faeces contaminated environment | No | 5 (8.06) | 1 | 1 | **0.000*** |
| | Yes | 85 (24.36) | 4.8280 (1.8866, 12.3553) | 5.8657 (2.2152, 15.5317) | |

COR, Crude odds ratio; AOR, Adjusted odds ratio; 1, Reference

*, Significant variables ($P < 0.05$)

**Table 7. Coliform bacterial isolates resistant to each antibiotic tested.**

| Antibiotic disc (potency) | Number of resistant isolates (%) | | | | | | | | | |
|---|---|---|---|---|---|---|---|---|---|---|
| | *Citrobacter freudii* (n = 2) | *Enterobacter cloacae* (n = 127) | *Enterobacter sakazakii* (n = 10) | *Escherichiacoli* (n = 37) | *Klebsiella pneumoniae* (n = 15) | *Klebsiella oxytoca* (n = 10) | *Serratia ficaria* (n = 2) | *Serratia liquefaciens* (n = 1) | *Serratia odorifera* (n = 1) | Total (n = 205) |
| KF (30μg) | 1 (50.0) | 107 (84.3) | 10 (100.0) | 27 (73.0) | 4 (26.7) | 1 (10.0) | 2 (100.0) | 1 (100.0) | 1 (100.0) | 154 (75.1) |
| AMX (10μg) | 2 (100.0) | 116 (91.3) | 10 (100.0) | 26 (70.3) | 15 (100.0) | 9 (90.0) | 2 (100.0) | 1(100.0) | 1(100.0) | 182 (88.8) |
| AMP (10μg) | 2 (100.0) | 66 (52.0) | 4 (40.0) | 19 (51.4) | 14 (93.3) | 8 (80.0) | 2 (100.0) | 1 (100.0) | 1 (100.0) | 117 (57.1) |
| NOR (5 μg) | 0 (0.0) | 2 (1.6) | 1 (10.0) | 4 (10.8) | 0 (0.0) | 0 (0.0) | 0 (0.0) | 0 (0.0) | 0 (0.0) | 7 (3.4) |
| NAL (30μg) | 0 (0.0) | 19 (15.0) | 1 (10.0) | 7 (18.9) | 0 (0.0) | 1 (10.0) | 0 (0.0) | 0 (0.0) | 0 (0.0) | 28 (13.8) |
| GEN (10μg) | 0 (0.0) | 18 (14.2) | 0 (0.0) | 1 (2.7) | 0 (0.0) | 0 (0.0) | 1 (50.0) | 1 (100.0) | 0 (100.0) | 21 (10.2) |
| STR (10μg) | 0 (0.0) | 29 (22.8) | 0 (0.0) | 18 (48.6) | 11 (73.3) | 1 (10.0) | 1 (50.0) | 0 (0.0) | 1 (100.0) | 61 (29.8) |
| TET (30μg) | 2 (100.0) | 27 (21.3) | 3 (30.0) | 12 (32.4) | 7 (46.7) | 0 (0.0) | 2 (100.0) | 1 (100.0) | 1 (100.0) | 55 (26.8) |
| CHL (10μg) | 1 (50.0) | 47 (37.0) | 4 (40.0) | 7 (18.9) | 4 (26.7) | 0 (0.0) | 2 (100.0) | 0 (0.0) | 0 (0.0) | 65 (31.7) |
| ERY (30μg) | 1 (50.0) | 65 (51.2) | 5 (50.0) | 25 (67.6) | 15 (100.0) | 10 (10.0) | 2 (100.0) | 1 (100.0) | 1 (100.0) | 125 (61.0) |
| COT (25μg) | 0 (0.0) | 16 (12.6) | 0 (0.0) | 13 (35.1) | 5 (33.3) | 1 (10.0) | 2 (100.0) | 1 (100.0) | 1 (100.0) | 39 (19.0) |

KF, Cephalothin,; AMX, Amoxicillin; AMP, Ampicillin; NOR, Norfloxacin; NAL, Nalidixic acid; GEN, Gentamicin; STR, Streptomycin; TET, Tetracycline; CHL, Chloramphenicol; ERY, Erythromycin; COT, Cotrimoxazole; μg, micrograms

205) *Enterobacter cloacae*, 10.7% (22/205) *E. coli*, 6.3% (13/205) *Klebsiella pneumoniae*, 2.9% (6/205) *Enterobacter sakazakii*, 2.0% (4/205) *Klebsiella oxytoca*, 1.0% (2/205) *Citrobacter freundii*, 1.0% (2/205) *Serratia ficaria*, 0.5% (1/205) *Serratia liquefaciens* and 0.5% (1/205) *Serratia odorifera*) (Fig 4). All isolates (100.0%) of *Citrobacter freundii* and *Serratia* spp., 68.0% *Klebsiella* spp., 59.5% *E. coli* and 46.0% *Enterobacter* spp. were MDR.

## 4. Discussion

This is the first report of the presence and prevalence of bovine mastitis (a worldwide dairy animal disease associated with significant economic losses) in the North West region of

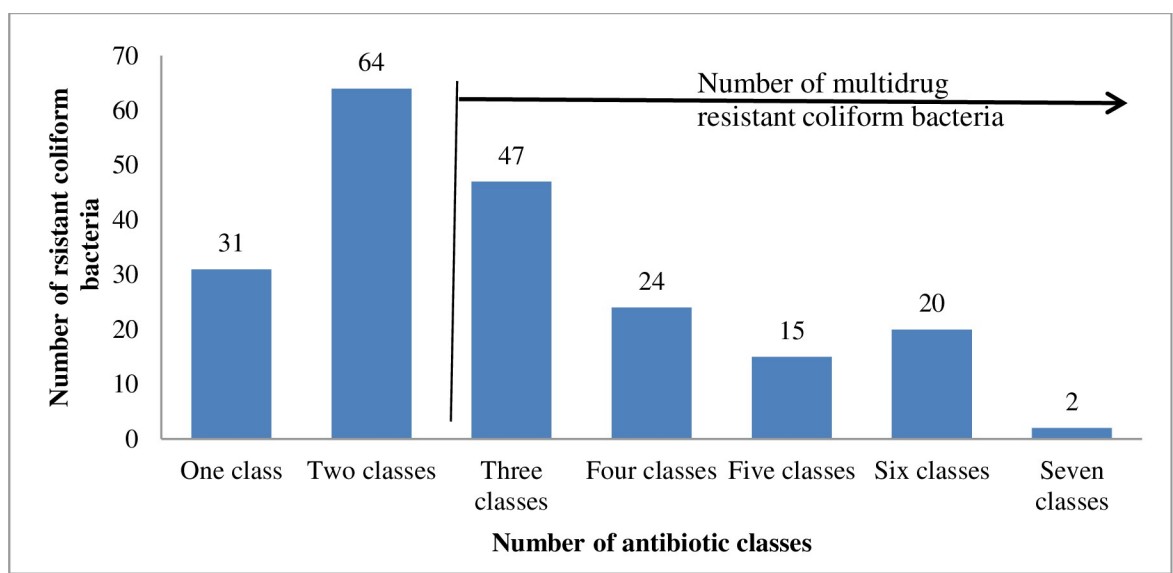

**Fig 4. Number of resistant coliform bacterial isolates against number of antibiotic classes.**

Cameroon. The California mastitis test (CMT) is an on-farm screening test in which the degree of gel formation is scored to estimate somatic cell count in milk samples and thus detect mastitis cases. The CMT is most helpful in detecting subclinical mastitis but serves little use in detecting clinical mastitis, although accurate [33]. This is because the presence of clinical signs of mastitis establishes a diagnosis of clinical mastitis. Bacterial culture remains the gold standard for confirming mastitis caused by microorganisms [34].

According to this cross-sectional study, the overall prevalence of mastitis among lactating dairy cows as determined by CMT and clinical examination was 53.0%. This finding is relatively lower compared to reports in other parts of the country [35] and elsewhere in Africa [36, 37]. However, our finding is relatively higher compared with reports from other countries in Africa [38–40] and out of Africa [41]. This variability in the prevalence of mastitis in different reports could suggest the complexity of the disease. According to Radostits et al. [9], the prevalence of bovine mastitis is expected to vary from place to place with the interaction of several factors, including herd management, environment and cow-related factors.

As expected, the prevalence of subclinical mastitis was higher (45.0%) than that of clinical mastitis (8.0%). Similar reports of subclinical mastitis dominance over clinical mastitis have been reported in several studies [35, 37, 40]. According to Sori et al. [23], subclinical mastitis was higher than clinical mastitis because, in most cases of infection, the cow mounts defence mechanisms in the udder, which reduces the severity of the disease. Another reason is the unawareness of farmers about subclinical cases of mastitis since symptoms are not evident [42], such that it is not diagnosed early and treated. Hence, it is important to educate farmers particularly about subclinical mastitis.

The prevalence of coliform-associated mastitis among cows in this study was 21.9% and this was higher than 8.8% in Nigeria [43] and 7.2% in Rwanda [44]. The high prevalence may indicate contamination from soil and faecal matter. Coliform bacteria originate from the cow's environment, such as faecal material, contaminated bedding, water and cow body sites [8]. They are generally acquired from the environment between milking through the teat canal into the udder when teat-ends contact an environmental site contaminated with coliform organisms [45]. Therefore, improving hygiene and reducing exposure of teat ends to environmental contamination is of paramount importance.

In this study, coliform bacteria were isolated in 12.8% of all quarter milk samples, which is relatively lower compared with 14.4% reported by Byarugaba *et al.* [46] in Uganda. The occurrence of coliform bacteria in all mastitis quarters was 24.8%, and this was lower compared to 66.0% in Tanzania [22] as well as 31.9% in Jordan [47]. The coliform bacteria isolated from mastitis milk samples were *Enterobacter cloacae* (12.6%), *Escherichia coli* (7.0%), *Klebsiella pneumoniae* (2.4%), *Enterobacter sakazakii* (1.1%), *Klebsiella oxytoca* (0.8%), *Citrobacter freudii* (0.4%), *Serratia ficaria* (0.4%), *Serratia liquefaciens* (0.2%), and this corroborates Hogan and Smith [8], who reported that these genera of coliform bacteria were frequently isolated from bovine mastitis cases. In their studies, Ahmed and Shimamoto [48] in Egypt and Kateete et al. [18] in Kampala-Uganda isolated these coliform genera, with *Escherichia coli* being the predominant coliform. Ngwa *et al.* [35] in Adamawa—Cameroon, reported *E. coli* as the predominant among coliforms associated with mastitis,. Makolo et al. [43] and Mbuk et al. [49] in Nigeria reported *Klebsiella* spp. as the predominant coliform. The differences in the relative occurrence of coliform bacteria could be due to differences in bacterial load of the various coliforms in the various environmental sources.

Mastitis is a complex disease influenced by several factors [9] and identification of risk factors in an area is important for the design of control programs [50]. Among the risk factors assessed, this study revealed a significant association ($P < 0.05$) of coliform mastitis prevalence

with lactation stage, breed, clinical history and moist/muddy faeces contaminated environment.

The association of coliform mastitis with lactation stage was reported in previous studies [51, 52]. Cows in early lactation were four times more likely to have coliform mastitis than cows in mid-lactation. This could be due to impaired immune function in early lactation related to the stress of producing a high amount of milk [53]. It could also be due to the absence of a dry cow therapy regime. During the dry period, pathogens penetrate the teat canal from the cow environment, multiply, and this can be carried over to the post parturient period and ultimately cause mastitis [54].

The present finding of an association of coliform mastitis with a history of mastitis was in harmony with other reports [55–57]. Cows with a history of mastitis were two times more likely to have coliform mastitis than those with no history. The current result may imply that the treatment of cows for mastitis may not be effective in eradicating the pathogens [57]. It could also be due to repeated challenges of the mammary tissues with coliforms coupled with other stress factors resulting in more significant risks of re-infection from the environment [54].

As revealed in this study, Oliveira et al. [57] and Taponen et al. [58] also reported that breed was a significant risk factor of coliform mastitis. Exotic-local cross breeds and Holstein-Friesian breeds were 2.5 times and 1.3 times likely to have coliform mastitis than the local African breeds. The occurrence of mastitis is generally higher in high milk-yielding cows than low-yielding cows [50, 59] because the high-yielding cows may be associated with a looser teat canal and milk leaking tendency, which predispose the udder to coliform bacterial invasion via the teat opening [57].

Moisture, mud and manure present in the environment of the animals are primary sources of exposure for environmental mastitis pathogens [21], which agreed with this study as cows in moist/muddy faeces contaminated environments were 5.9 times more likely to have coliform mastitis than those in a clean environment.

Although this study did not show age and parity as significantly associated with coliform mastitis in cows, Hogan and Smith [8] reported increase susceptibility to coliform mastitis with an increase in age and parity. It was observed that farmers in this study did not keep records, so data on age and parity may not be accurate, particularly in cases where cows were brought from elsewhere into a herd.

*In vitro*, antibiotic susceptibility testing was performed on coliform isolates from both mastitis positive and negative quarters. Coliform bacterial isolates (88.8%) exhibited resistance against amoxicillin, although Mbuk et al. [49] reported no resistance. In our study, 75.1% and 57.1% of coliform isolates were resistant to cephalothin and ampicillin, respectively; Kateete et al. [18] reported a lower resistance of 33.0% for cephalothin and higher resistance of 71.0% for ampicillin in Uganda. Worldwide, the β-lactam class of antibiotics is the most commonly used in human and veterinary medicine [60, 61]. This may explain the reports of high resistance of coliform bacteria against them. This study found that 61.0% isolates were resistant to erythromycin; Mbuk et al. [49] and Makolo et al. [43] had more than 75.0% resistance. The use of chloramphenicol in food-producing animals in many countries has been prohibited to avoid the danger of resistance in human medicine [62]. Resistance against chloramphenicol was exhibited by 31.7% of coliform isolates in this study similar to studies elsewhere [43, 49].

This study revealed that norfloxacin (3.4%), gentamicin (10.2%), nalidixic acid (13.8%) and cotrimoxazole (19.0%) were the antibiotics with the least resistance by coliforms. Similar results of low resistance of coliform isolates to these antibiotics have been reported elsewhere [43, 49, 63]. The least resistance by norfloxacin in our study agrees with Wenz et al. [64] who

recommended fluoroquinolones as the drugs of choice for mastitis caused by Gram-negative rods.

Generally, although coliform bacteria exhibited low resistance to some of the antibiotics tested and could be recommended as a drug of choice for coliform-associated mastitis, the pattern of resistance differed with specific coliform genera. According to World Health Organization reports [65], the resistance of *Escherichia coli* to fluoroquinolones is pervasive in a range of 0.0–98.0% in Africa. For example, in this study, resistance against nalidixic acid and norfloxacin was 18.9% and 10.8% for *E. coli* strains, respectively (Table 7). Resistance to gentamicin and cotrimoxazole was exhibited by 50.0% and 100.0% of *Serratia* spp. respectively. Thus, it is important to isolate the coliform mastitis pathogen and perform an antibiotic susceptibility test, if possible, before any antibiotic therapy.

Though the choice of antibiotic for mastitis treatment depended on the veterinarian in the study area, it was observed that the two most commonly used drugs for mastitis treatment were penicillin-streptomycin and tetracyclines. This study revealed that *Klebsiella pneumoniae* (46.7%), *E. coli* (32.4%), *Enterobacter sakazakii* (30.0%) and *Enterobacter cloacae* (21.3%) isolates were resistant to tetracycline. Mbuk et al. [49] reported more than 50.0% resistance to tetracycline by *Klebsiella* spp. and *Enterobacter* spp.. Makolo et al. [43] reported 100.0% resistance of *E. coli* isolates to tetracycline. Resistance to streptomycin was exhibited by 73.3% *Klebsiella pneumoniae*, 48.6% *E. coli*, 22.8% *Enterobacter cloacae* and 10.0% *Klebsiella oxytoca*. Resistance of coliforms to streptomycin conforms to the report by Makolo *et al.* [43], who revealed 100.0% resistance exhibited by *Klebsiella pneumoniae* and *E. coli* isolates each. Thus, it is not advisable to use these antibiotics to treat coliform-associated bovine mastitis without performing an antibiotic susceptibility test.

Analysis of MDR among coliform bacterial isolates revealed 52.7% (108/205) in the proportion of *Enterobacter* spp. (63 isolates, 30.7%), *E. coli* (22 isolates, 10.7%), *Klebsiella* spp. (17 isolates, 8.3%), *Serratia* spp. (4 isolates, 2.0%) and *Citrobacter freundii* (2 isolates, 1.0%). Similarly, Ahmed and Shimamoto [48] in Egypt reported 27.8% MDR among coliforms in the proportion of *Klebsiella* spp. (14 isolates, 12.6%), *Enterobacter* spp. (8 isolates, 7.1%), *E. coli* (5 isolates, 4.5%), *Citrobacter freundii* (3 isolates, 2.7%) and *Serratia* sp. (1 isolate, 0.9%). The presence of MDR coliform bacteria in milk is a serious cause for concern, particularly isolates exhibiting resistance to as many as seven classes of antibiotics like the two *Enterobacter cloacae* isolates in this study. The great differences observed in the antibiotic resistance of coliform bacteria from different studies indicate the importance of antibiotic susceptibility tests and periodic surveillance of the antibiotic susceptibilities associated with mastitis.

## 5. Conclusion

Based on the data obtained in this study area, the prevalence of mastitis among lactating cows was 53.0%, comprising 45.0% (185/411) subclinical and 8.0% (33/411) clinical cases. The prevalence of coliform-associated mastitis at the cow-level was 21.9% (90/411). Coliform bacteria were isolated from 12.8% (205/1608) of all quarter milk samples while 24.8% (132/532) of all mastitis quarters were positive for coliforms: *Enterobacter* cloacae (12.6%), *Escherichia coli* (7.0%), *Klebsiella pneumoniae* (2.4%), *Enterobacter sakazakii* (1.1%), *Klebsiella oxytoca* (0.8%), *Citrobacter freundii* (0.4%), *Serratia ficaria* (0.4%) and *Serratia liquefaciens* (0.2%). Prevalence of coliform mastitis was significantly ($P < 0.05$) associated with lactation stage, cow breed, history of mastitis and moist/muddy faeces contaminated environment. Amoxicillin had the least activity against coliform bacteria, and norfloxacin was the most active antibiotic. MDR was observed in 52.7% (108/205) of the coliform isolates.

Cross-sectional studies like this can only give associations but not causality, so an experimental study in this area will elucidate these associations as actual causes of coliform mastitis. However, the presence of mastitis, particularly coliform-associated mastitis, warrants the application of good hygiene practices in the milking process and the cow's environment. Raising awareness of mastitis (especially subclinical mastitis) and the non-misuse of antibiotics among farmers through extension services and restricting consumption of unpasteurized milk is vital in improving cow and public health. Antibiotic resistance monitoring should also be implemented.

## Acknowledgments

We sincerely thank all the milk-producing farmers, the Regional Delegate and staff of the Delegation of Livestock, Fisheries and Animal Industries of the North West Region for facilitating sample collection. The authors are grateful to the Laboratory for Emerging Infectious Diseases, University of Buea, for providing equipment to accomplish this work.

## Author Contributions

**Conceptualization:** Ursula Anneh Abegewi, Roland N. Ndip, Lucy M. Ndip.

**Data curation:** Ursula Anneh Abegewi, Seraphine Nkie Esemu.

**Formal analysis:** Ursula Anneh Abegewi, Seraphine Nkie Esemu.

**Investigation:** Ursula Anneh Abegewi, Seraphine Nkie Esemu.

**Methodology:** Ursula Anneh Abegewi, Roland N. Ndip, Lucy M. Ndip.

**Resources:** Ursula Anneh Abegewi, Lucy M. Ndip.

**Supervision:** Seraphine Nkie Esemu, Lucy M. Ndip.

**Validation:** Roland N. Ndip, Lucy M. Ndip.

**Writing – original draft:** Ursula Anneh Abegewi, Seraphine Nkie Esemu.

**Writing – review & editing:** Roland N. Ndip, Lucy M. Ndip.

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
