## [Decision Letter · Decision Letter 0]

9 Dec 2021

PONE-D-21-33571Prevalence, Risk Factors and Antibiotic Resistance of Coliforms from Lactating Dairy Mastitis Cows in North West CameroonPLOS ONE

Dear Dr. Ndip,

Thank you for submitting your manuscript to PLOS ONE. After careful consideration, we feel that it has merit but does not fully meet PLOS ONE’s publication criteria as it currently stands. Therefore, we invite you to submit a revised version of the manuscript that addresses the points raised during the review process. In addition to dealing with comments from the two reviewers, I will ask you to revise the writing of legends which, as the manuscript stands, are not informative at all. I also encourage you to take into account the comment on the suitability of chosen antibiotics for coliforms. Furthermore, there seems to be a miscalculation in the number of coliform positive quarters (131 in the text and tables while, from figure 3, the calculated number is 55+(28*2)+(6*3)+(1*4)=133).

We look forward to receiving your revised manuscript.

Kind regards,

Pierre Germon

Academic Editor

PLOS ONE

Journal Requirements:

Reviewers' comments:

Reviewer's Responses to Questions

**Comments to the Author**

1. Is the manuscript technically sound, and do the data support the conclusions?

Reviewer #1: Partly

Reviewer #2: No

2. Has the statistical analysis been performed appropriately and rigorously? 

Reviewer #1: Yes

Reviewer #2: I Don't Know

3. Have the authors made all data underlying the findings in their manuscript fully available?

Reviewer #1: Yes

Reviewer #2: No

4. Is the manuscript presented in an intelligible fashion and written in standard English?

Reviewer #1: Yes

Reviewer #2: No

5. Review Comments to the Author

Reviewer #1: Review of:

Prevalence, Risk Factors and Antibiotic Resistance of Coliforms from Lactating Dairy Mastitis Cows in North West Cameroon

Comments:

L 1 The title of this manuscript is misleading. Clinical and subclinical infections of the mammary glands differ; whereas milk from cows with clinical mastitis is not approved for sale or consumption, but cows with subclinical mastitis may have their milk sold or consumed, depending on the Somatic Cell Count, which differs among countries. For example, in the EU, the SCC of milk for sale cannot exceed 400,000 over a 3-month period in herds that must be sampled at least once monthly. A more accurate title would simply delete the word “Mastitis” and discuss it in appropriate context in the manuscript.

L30 California Mastitis Test instead of California mastitis Test

L35 It is important here to define what is meant by the term “mastitis”. In this study, cows with active coliform organisms in their milk were classified as having mastitis, regardless of their SCC. Some of these cows would have met the European standards for salable milk. This needs to be clarified.

L37 It would be more correct to say that these cows had Coliform infections, rather than saying Coliform mastitis.

L81-82 This statement is incorrect. If only a few Coliform organisms were detected in milk from a quarter, then this would not be considered mastitis. It would be more correct to say that about 70-80% of clinical mastitis cases are associated with Coliform infections.

L88 There are several methods to treat Coliform infections instead of using antibiotics. For example, frequent milking to expel the infected milk, sometimes with use of oxytocin, can eliminate the infection without antibiotics.

L91-93 Transmission through the food chain can be reduced greatly by pasteurization of the milk.

L93-94 Monitoring resistance has no effect on control of mastitis unless cows with resistant Coliforms are culled.

L115 See this similar paper from Cameroon: https://www.heraldopenaccess.us/openaccess/bacterial-pathogens-involved-in-bovine-mastitis-and-their-antibiotic-resistance-patterns-in-the-adamawa-region-of-cameroon This paper may be from a different region and it differs in some respects from the current paper, but it clearly shows the causes of clinical mastitis in Cameroon.

L136 Although this is an important area in Cameroon, it is a tiny part of the country, Africa, and the world. What makes this study unique and of interest to scientists worldwide?

L186 Ideally, the veterinarian should have changed gloves between each teat, not just each cow.

L223 Please report the precise statistical models used for these analyses.

L232 Describe how the cows’ teats were cleaned before milking and whether their teats were dipped with disinfectants after milking. Also, did the milkers wear gloves and change gloves between cows? If none of these procedures were followed, then the levels of infection would be greater than if recommended milking procedures were used. Was the milking procedure recorded for each farm?

L234 What was the average lactation number for the cows? Among these herds, what was the average stage of lactation at sampling?

L252 ….13 out of 1644 quarters (2.2%) were….

L253-254 ….were blind, and 5.8% of the 1608 functional quarters (within 33 cows) displayed clinical signs characterized by watery milk or clots, flakes or blood.

L261 This entire section and tables are not useful or informative. There were too few cows per Division to be meaningful. The information may be useful for local farmers, but it is not useful for the scientific community at large.

L280 The term “subclinical mastitis” is not clearly defined by NMC or other organizations. Somatic cell count of milk that is acceptable for processing for human consumption differs among countries and even among processors within countries. Clearly, cows with extended periods of SCC above 100,000 produce less milk, but these cows are not classified as having mastitis unless the milk is abnormal or the udder is swollen, produces bloody milk or milk clots. It is more meaningful to look at the average SCC for the herd, but in this study, herd sizes are small, and a single cow can affect the average for the entire herd.

L280-onward Number of cows in each region is too small to be meaningful to the scientific community. This is useful information for the regions, but it is not useful for dairy farms elsewhere, because regions may differ in rates of infection.

L310 This section is useful and should be retained.

L350 This section is useful and should be retained.

L382 This section can be condensed to key points without making excessive comparisons among different countries. Also, include reports from dairy regions outside of Africa.

Reviewer #2: The paper describes investigations into the prevalence of Enterobacterales in milk samples from dairy cows in the North-West of Cameroon. As such this is a useful undertaking although the selective investigation of coliforms while neglecting other mastitis causing bacteria seems an odd approach. The very high prevalence of coliforms identified and the odd pattern with Enterobacter cloacae as the most frequent species shed doubt on the quality of sampling. Can the authors be sure, that their results truely reflect the situation in the udder and not (additional) environmental contamination of the samples with bacteria from the teat or udder surface or the surroundings? Interestingly, the authors never found a combination of different bacteria in the samples, which is a common issue in mastitis bacteriology. The authors should report, if they found the same bacteria in the udder quarters when several quarters were affected. Likewise they might discuss if certain bacteria were also clustered within herd, i.e. occuring in several cows in the same herd.

The authors spent a lot of room in the paper to the description of regional differences without providing evidence for the potential reasons for the differences. Including region in the statistical analysis might have helped to elucidate potential associations. For the international reader regions without a more detailed characterization of the differences between the regions are not useful.

The sample size calculation was done for independent entities. However, the authors report that they investigated 411 cows in only 123 farms. This means their sampling unit cow was not independent but clustered in farms. This is seemingly not refllected in the statistical analysis and -which is worse- neither included in the descriptive analysis nor in the discussion. An analysis of results also including herd size would be adequate.

Freezing milk samples prior to analysis usually is associated with a reduction in the number of coliform bacteria while it is not an issue with staphylococci. Therefore the high prevalence of coliforms is even more astonishing.

The antimicrobial test panel includes penicillin. This is not useful as coliforms are intrinsically resistant to penicillin. Therefore penicillin needs to be removed from the analysis and the analysis needs to be re-done. This also applies to the analysis of MDR as it is not valid if it includes penicillin.

When presenting the results of disk diffusion the authors need to present the used cut-off values. Ideally they would even provide the inhibition zone diameters.

Table 1 can be omitted as the information is included in table 8.

Tables 2 to 4 can be combined

Results in table 5 are odd as the number of cows is lower than in the other tables.

Only one decimal should be presented throughout.

In the discussion section excessive repetition of results should be avoided. Likewise comparison with other studies, while relevant should be condensed. Can the condition e.g. in Ethiopia or Tansania be compared with Cameroon?

A higher prevalence of subclinical as compared to clinical mastitis does not have to be elaborated as this is textbook knowledge.

The discussion on AMR cannot be evaluated as the results are not valid.

6. PLOS authors have the option to publish the peer review history of their article (what does this mean?). If published, this will include your full peer review and any attached files.

Reviewer #1: No

Reviewer #2: No

---

## [Author Response · Author response to Decision Letter 0]

9 Feb 2022

Point-by-point response letter

Dear Editor,

We appreciate your comments and those of the reviewers. We hope we have provided satisfactory responses/amendments to all the comments. The Point-by-point responses/ amendments are presented below. 

Editor’s comments

Editor: We invite you to submit a revised version of the manuscript that addresses the points raised during the review process

Response: We have submitted a revised version of the manuscript addressing the points raised during the review process. Thank you for this opportunity. Hope we have addressed the points satisfactorily.

…………………………………………………………………………………………………

Editor: I will ask you to revise the writing of legends which, as the manuscript stands, are not informative at all.

Response: We have revised the writing of legends to make them more informative for all Tables except Table 6. We have also revised the legends for all Figures. 

Table 1 (see L223-224….. New Table

Table 2 (see L280-281) ….. New Table

Table 3 (see L294)

Table 4 (see L298)

Table 5 (see L328)

Table 6 (see L333)

Table 7 (see L348)

Fig. 1 (see L137)

Fig. 2 (see L261)

Fig. 3 (see L310)

Fig. 4 (see L361-362)

…………………………………………………………………………………………………

Editor: I also encourage you to take into account the comment on the suitability of chosen antibiotics for coliforms.

Response: We have removed penicillin from the antibiotics panel and re-analyzed the results of the antibiotic susceptibility testing. 

…………………………………………………………………………………………………

Editor: There seems to be a miscalculation in the number of coliform positive quarters (131 in the text and tables while, from figure 3, the calculated number is 55+(28*2)+(6*3)+(1*4)=133)

Response: We have done the correction (see Fig 3– L310). Actually, the number of coliform-associated mastitis quarters is 132. Thank you very much.

…………………………………………………………………………………………………

Editor: Please include the following items when submitting your revised manuscript:

Response: We have submitted all items requested above.

…………………………………………………………………………………………………

Reviewer #1 comments

Reviewer #1: L1 The title of this manuscript is misleading. Clinical and subclinical infections of the mammary glands differ; whereas milk from cows with clinical mastitis is not approved for sale or consumption, but cows with subclinical mastitis may have their milk sold or consumed, depending on the Somatic Cell Count, which differs among countries. For example, in the EU, the SCC of milk for sale cannot exceed 400,000 over a 3-month period in herds that must be sampled at least once monthly. A more accurate title would simply delete the word “Mastitis” and discuss it in appropriate context in the manuscript.

Response: The title has been revised. See L1-2

…………………………………………………………………………………………………

Reviewer #1: L30 California Mastitis Test instead of California mastitis Test

Response: This has been corrected. See L30

…………………………………………………………………………………………………

Reviewer #1: L35 It is important here to define what is meant by the term “mastitis”. In this study, cows with active coliform organisms in their milk were classified as having mastitis, regardless of their SCC. Some of these cows would have met the European standards for salable milk. This needs to be clarified.

Response: This is not true! In this study, we did not classify ALL cows with active coliform organisms in their milk as having mastitis. Mastitis was detected by clinical examination and the California Mastitis Test (CMT) – and we considered CMT scores ≥1 (which give SCC estimates ≥ 400,000 cells/ml) as positive (See L173). In fact, we identified 73 quarters (within 38 cows) that had active coliform organisms but were not classified as mastitic (see Table 3 – Line 294).

…………………………………………………………………………………………………

Reviewer #1: L37 It would be more correct to say that these cows had Coliform infections, rather than saying Coliform mastitis.

Response: We maintain coliform mastitis (coliform-associated mastitis) because, in this study, we detected mastitis by clinical examination and the California Mastitis Test (CMT). Mastitis positive cows or udder quarters that had coliform bacteria isolated from their milk were considered positive for coliform mastitis. While a total of 205/1608 (12.8%) quarter milk samples had active coliform bacteria, only 132 of these samples were from mastitic cows. Hence, coliform infection is not synonymous to coliform mastitis.

…………………………………………………………………………………………………Reviewer #1: L81-82 This statement is incorrect. If only a few Coliform organisms were detected in milk from a quarter, then this would not be considered mastitis. It would be more correct to say that about 70-80% of clinical mastitis cases are associated with Coliform infections.

Response: You are very right! The whole sentence has been deleted. …………………………………………………………………………………………………

Reviewer #1: L88 There are several methods to treat Coliform infections instead of using antibiotics. For example, frequent milking to expel the infected milk, sometimes with use of oxytocin, can eliminate the infection without antibiotics.

Response: This is true. We have revised the sentence to indicate that antibiotic therapy is not the only method to treat coliform infections (See L87-90)

…………………………………………………………………………………………………

Reviewer #1: L91-93 Transmission through the food chain can be reduced greatly by pasteurization of the milk.

Response: True. For clarity, we have rephrased the sentence by replacing food chain with unpasteurized milk (See L91-93)

…………………………………………………………………………………………………

Reviewer #1: L93-94 Monitoring resistance has no effect on control of mastitis unless cows with resistant Coliforms are culled.

Response: We think monitoring antibiotic resistance has an effect on the control of mastitis even when cows with resistant coliforms are not culled. To substantiate, we quote this “Therefore, continuous monitoring of antimicrobial resistance (AMR) and application of AMR mitigation measures are required to control their spread to humans, animals, and the environment” (Abdi et al. 2021; https://doi.org/10.3390/ani11010131). However, for clarity, we have revised this sentence. See L94-95

…………………………………………………………………………………………………

Reviewer #1: L115 See this similar paper from Cameroon: https://www.heraldopenaccess.us/openaccess/bacterial-pathogens-involved-in-bovine-mastitis-and-their-antibiotic-resistance-patterns-in-the-adamawa-region-of-cameroon. This paper may be from a different region and it differs in some respects from the current paper, but it clearly shows the causes of clinical mastitis in Cameroon.

Response: We have read the paper! Yes, this paper identifies 14 bacterial species as causes of mastitis (clinical and subclinical) in Cameroon. Similarly, in our study, we have identified nine coliform bacterial species including two species (Escherichia coli and Klebsiella pneumoniae reported in the previous study) and seven species reported only in our study. This means that our study has expanded the repertoire of bacterial isolates associated with mastitis in Cameroon. Hence, the results of this study have provided important epidemiological data on coliform-associated mastitis.

…………………………………………………………………………………………………

Reviewer #1: L136 Although this is an important area in Cameroon, it is a tiny part of the country, Africa, and the world. What makes this study unique and of interest to scientists worldwide?

Response: This study is unique, in that, it has expanded knowledge on the epidemiology of mastitis in Cameroon in the following ways: 

It has expanded the repertoire of bacterial isolates associated with mastitis in Cameroon. Bacterial isolates associated with mastitis may vary from herd to herd as well as within and between countries. Hence, the prompt identification and understanding of the diversity of the pathogens associated with mastitis is essential for effective control (Pascu et al., 2022; https://doi.org/10.3390/antibiotics11010057).

It has identified some risk factors of coliform-associated mastitis.

It has reported multidrug resistance of coliform bacteria in milk 

Although our study area is a tiny part of the country, Africa, and the world, it has contributed data to the body of knowledge of bovine mastitis in Cameroon. Such knowledge is of interest to scientists worldwide seeking information from Cameroon on such a topic, which as at now is scanty in literature. Thus, the uniqueness and interest of this study from this tiny area to scientists worldwide cannot be overemphasized.

…………………………………………………………………………………………………

Reviewer #1: L186 Ideally, the veterinarian should have changed gloves between each teat, not just each cow.

Response: We have noted this remarked and thank you very much. However, the veterinarian disinfected gloves with 70% alcohol between each teat.

…………………………………………………………………………………………………

Reviewer #1: L223 Please report the precise statistical models used for these analyses.

Response: We have reported the precise statistical models used for these analyses (See L226)

…………………………………………………………………………………………………

Reviewer #1: L232 Describe how the cows’ teats were cleaned before milking and whether their teats were dipped with disinfectants after milking. Also, did the milkers wear gloves and change gloves between cows? If none of these procedures were followed, then the levels of infection would be greater than if recommended milking procedures were used. Was the milking procedure recorded for each farm?

Response: Yes, we observed the milking procedures in the farms and these procedures have been reported (See L243-248).

…………………………………………………………………………………………………

Reviewer #1: L234 What was the average lactation number for the cows? Among these herds, what was the average stage of lactation at sampling?

Response: We have included these details (See L249-250).

…………………………………………………………………………………………………

Reviewer #1: L252 ….13 out of 1644 quarters (2.2%) were….

Response: We have effected this correction (See L253-254)

…………………………………………………………………………………………………

Reviewer #1: L253-254 ….were blind, and 5.8% of the 1608 functional quarters (within 33 cows) displayed clinical signs characterized by watery milk or clots, flakes or blood.

Response: We have effected this correction (See L254- 255)

…………………………………………………………………………………………………

Reviewer #1: L261 This entire section and tables are not useful or informative. There were too few cows per Division to be meaningful. The information may be useful for local farmers, but it is not useful for the scientific community at large.

Response: Okay. We have modified this entire section to capture prevalence in the region (i.e, the North West region) without the divisions. See L263.

…………………………………………………………………………………………………

Reviewer #1: L280 The term “subclinical mastitis” is not clearly defined by NMC or other organizations. Somatic cell count of milk that is acceptable for processing for human consumption differs among countries and even among processors within countries. Clearly, cows with extended periods of SCC above 100,000 produce less milk, but these cows are not classified as having mastitis unless the milk is abnormal or the udder is swollen, produces bloody milk or milk clots. It is more meaningful to look at the average SCC for the herd, but in this study, herd sizes are small, and a single cow can affect the average for the entire herd.

Response: This remark is noted! A CMT score of ≥1 is expected to have an SCC of ≥400,000 according to the CMT kit we used. Since this was a cross-sectional study, SCC over an extended period could not be determined. 

…………………………………………………………………………………………………

Reviewer #1: L280-onward Number of cows in each region is too small to be meaningful to the scientific community. This is useful information for the regions, but it is not useful for dairy farms elsewhere, because regions may differ in rates of infection.

Response: We have removed the results of the divisions from this section and we have provided the results of the prevalence of coliform-associated mastitis in the North West region in Table 2 (See L280-281)

…………………………………………………………………………………………………

Reviewer #1: L310 This section is useful and should be retained.

Response: Okay.

…………………………………………………………………………………………………

Reviewer #1: L350 This section is useful and should be retained.

Response: Okay.

Reviewer #1: L382 This section can be condensed to key points without making excessive comparisons among different countries. Also, include reports from dairy regions outside of Africa.

Response: We have revised this section (see L365). Thank you very much. 

…………………………………………………………………………………………………

Reviewer #2 comments

Reviewer #2: The paper describes investigations into the prevalence of Enterobacterales in milk samples from dairy cows in the North-West of Cameroon. As such this is a useful undertaking although the selective investigation of coliforms while neglecting other mastitis causing bacteria seems an odd approach. The very high prevalence of coliforms identified and the odd pattern with Enterobacter cloacae as the most frequent species shed doubt on the quality of sampling. Can the authors be sure, that their results truly reflect the situation in the udder and not (additional) environmental contamination of the samples with bacteria from the teat or udder surface or the surroundings? Interestingly, the authors never found a combination of different bacteria in the samples, which is a common issue in mastitis bacteriology. The authors should report, if they found the same bacteria in the udder quarters when several quarters were affected. Likewise they might discuss if certain bacteria were also clustered within herd, i.e. occurring in several cows in the same herd.

Response: 

- We followed aseptic measures during sample collection (See L183). So we are certain the results truly reflect the situation in the udder and not from the outside of the cow. 

- We have reported the coliform bacteria type isolated from mastitis udder quarters when several quarters were affected (See L297-302). However, the same bacteria were identified in the udder quarters when several quarters were affected in majority of the cows (see L302 – 307). 

…………………………………………………………………………………………………

Reviewer #2: The authors spent a lot of room in the paper to the description of regional differences without providing evidence for the potential reasons for the differences. Including region in the statistical analysis might have helped to elucidate potential associations. For the international reader regions without a more detailed characterization of the differences between the regions are not useful.

Response: Thank you for this enlightenment. We have removed the results for the divisions since Reviewer 1 raised the same issue and requested we do not report. We have, therefore, reported only the overall prevalence for the North West region (see Table 2, L280-281).

…………………………………………………………………………………………………

Reviewer #2: The sample size calculation was done for independent entities. However, the authors report that they investigated 411 cows in only 123 farms. This means their sampling unit cow was not independent but clustered in farms. This is seemingly not reflected in the statistical analysis and -which is worse- neither included in the descriptive analysis nor in the discussion. An analysis of results also including herd size would be adequate.

Response: Yes, in this study cows were considered independent entities. We have now added herd size in the analysis of the results (See Table 5, L328)

…………………………………………………………………………………………………

Reviewer #2: Freezing milk samples prior to analysis usually is associated with a reduction in the number of coliform bacteria while it is not an issue with staphylococci. Therefore the high prevalence of coliforms is even more astonishing.

Response: True, freezing milk samples before analysis is usually associated with a reduction in the number of coliform bacteria. However, we did not investigate the number of coliform bacteria in this study to be able to notice the effect of freezing. 

…………………………………………………………………………………………………

Reviewer #2: The antimicrobial test panel includes penicillin. This is not useful as coliforms are intrinsically resistant to penicillin. Therefore penicillin needs to be removed from the analysis and the analysis needs to be re-done. This also applies to the analysis of MDR as it is not valid if it includes penicillin.

Response: We have removed penicillin from the antimicrobial test panel and the analysis has been re-done.

………………………………………………………………………………………………….

Reviewer #2: When presenting the results of disk diffusion the authors need to present the used cut-off values. Ideally they would even provide the inhibition zone diameters.

Response: We have provided a table of zone diameter breakpoints of the various antibiotics against coliforms (See Table 1, L223-224)

…………………………………………………………………………………………………

Reviewer #2: Table 1 can be omitted as the information is included in table 8.

Response: This has been done.

…………………………………………………………………………………………………

Reviewer #2: Tables 2 to 4 can be combined

Response: We have combined the tables as one. But we revised the table to capture results of mastitis prevalence in the North West region without divisional prevalence. See L263.

…………………………………………………………………………………………………

Reviewer #2: Results in table 5 are odd as the number of cows is lower than in the other tables.

Response: Table 5 has been deleted but the information has been maintained in text. See L368-273.

…………………………………………………………………………………………………

Reviewer #2: Only one decimal should be presented throughout.

Response: This has been done. Thank you very much.

…………………………………………………………………………………………………

Reviewer #2: In the discussion section excessive repetition of results should be avoided. Likewise comparison with other studies, while relevant should be condensed. Can the condition e.g. in Ethiopia or Tanzania be compared with Cameroon?

Response: We have revised the discussion section, making sure there is no excessive repetition.

Ethiopia, for example, has one of the largest livestock populations in Africa and modern dairy farming is beginning to flourish in this country in both urban and peri-urban areas of the major towns. Several studies have been conducted in various parts of the country on bovine mastitis leading to increased knowledge on prevalence of mastitis, the microbial diversity and risk factors associated with disease development. Cameroon has a comparatively smaller livestock population and very scanty literature on the occurrence of bovine mastitis. 

However, these countries are in sub-Saharan Africa and the conditions in these countries may be more similar compared to countries elsewhere. Yet, the prevalence of bovine mastitis in these countries is expected to vary due to the interaction of several factors, including herd management, environment and cow-related factors.

……………………………………………………………………………………………

Reviewer #2: A higher prevalence of subclinical as compared to clinical mastitis does not have to be elaborated as this is textbook knowledge.

Response: In most sub-Saharan countries including Cameroon, subclinical mastitis has received little or no attention and efforts are focused on the treatment of clinical cases while high productive and economic losses come from subclinical mastitis. Therefore, highlighting this is very important for public awareness to guide mitigation. In fact, several different studies point out that subclinical mastitis is more economically important than clinical mastitis due to the fact that subclinical mastitis is more difficult to diagnose and therefore usually persists longer in the herds, causing production losses. 

…………………………………………………………………………………………………

Reviewer #2: The discussion on AMR cannot be evaluated as the results are not valid.

Response: The results are now valid because we have removed penicillin from the panel of antibiotics.

…………………………………………………………………………………………………

---

## [Decision Letter · Decision Letter 1]

13 Apr 2022

PONE-D-21-33571R1Prevalence and Risk Factors of Coliform-Associated Mastitis and Antibiotic Resistance of Coliforms from Lactating Dairy Cows in North West CameroonPLOS ONE

Dear Dr. Ndip,

Thank you for submitting your manuscript to PLOS ONE. After careful consideration, we feel that it has merit but does not fully meet PLOS ONE’s publication criteria as it currently stands. Therefore, we invite you to submit a revised version of the manuscript that addresses the points raised during the review process.

We look forward to receiving your revised manuscript.

Kind regards,

Pierre Germon

Academic Editor

PLOS ONE

Journal Requirements:

Reviewers' comments:

Reviewer's Responses to Questions

**Comments to the Author**

1. If the authors have adequately addressed your comments raised in a previous round of review and you feel that this manuscript is now acceptable for publication, you may indicate that here to bypass the “Comments to the Author” section, enter your conflict of interest statement in the “Confidential to Editor” section, and submit your "Accept" recommendation.

Reviewer #1: All comments have been addressed

Reviewer #3: (No Response)

2. Is the manuscript technically sound, and do the data support the conclusions?

Reviewer #1: Yes

Reviewer #3: Yes

3. Has the statistical analysis been performed appropriately and rigorously? 

Reviewer #1: Yes

Reviewer #3: Yes

4. Have the authors made all data underlying the findings in their manuscript fully available?

Reviewer #1: Yes

Reviewer #3: Yes

5. Is the manuscript presented in an intelligible fashion and written in standard English?

Reviewer #1: Yes

Reviewer #3: Yes

6. Review Comments to the Author

Reviewer #1: The authors have made recommended changes in the manuscript and have clarified issues from the initial review. This manuscript will be of interest to scientists and will add useful information of coliform resistance and on mastitis in this part of Africa.

Reviewer #3: I only have minor comments for this version.

Lines 477-490: when it comes to the discussion of percent resistant isolates, I think the authors should be much more cautious in their statements. What is the meaning of 100% resistant isolates for a particular antibiotic when only one or two strains of a species have been isolated ? Not much. If read as such, one could take as a general message that all Serratia spp. or Citrobacter freundii isolated from milk are resistant to tetracycline. Drawing such a conclusion from 1 or 2 isolates is really misleading. I think the authors should somehow nuance their remarks and focus on species for which they have at least 10 isolates.

Line 475: the statement "Thus, it is important to isolate the coliform mastitis pathogen and perform an antibiotic susceptibility test before any antibiotic therapy." is not always realistic when it comes to application on the field. Very often, when a clinical mastitis case is detected, antibiotics are used without any prior identification of the pathogen. I am not saying it is not important to follow antibiotic resistance of mastitis pathogens, I'm rather implying that it not easily feasible in the field and that it is more a question for surveillance authorities.This statement should be somehow modified.

7. PLOS authors have the option to publish the peer review history of their article (what does this mean?). If published, this will include your full peer review and any attached files.

Reviewer #1: No

Reviewer #3: No

---

## [Author Response · Author response to Decision Letter 1]

24 Apr 2022

Point-by-point response letter

Dear Editor,

We appreciate your comments and those of the reviewers. We hope we have provided satisfactory responses/amendments to all the comments. The Point-by-point responses/ amendments are presented below. 

Editor’s comments

Editor: We invite you to submit a revised version of the manuscript that addresses the points raised during the review process.

Response: We have submitted a revised version of the manuscript addressing the points raised during the review process. Thank you for this opportunity. Hope we have addressed the points satisfactorily.

…………………………………………………………………………………………………

Editor: Please include the following items when submitting your revised manuscript:

Response: We have submitted all items requested above.

…………………………………………………………………………………………………

Editor: Please review your reference list to ensure that it is complete and correct. If you have cited papers that have been retracted, please include the rationale for doing so in the manuscript text, or remove these references and replace them with relevant current references. Any changes to the reference list should be mentioned in the rebuttal letter that accompanies your revised manuscript. If you need to cite a retracted article, indicate the article’s retracted status in the References list and also include a citation and full reference for the retraction notice.

Response: We have reviewed the reference list to ensure it is complete and correct.

………………………………………………………………………………………………………

Editor: While revising your submission, please upload your figure files to the Preflight Analysis and Conversion Engine (PACE) digital diagnostic tool, https://pacev2.apexcovantage.com/.

Response: We have uploaded figure files to the PACE digital diagnostic tool to ensure they meet PLOS requirements.

………………………………………………………………………………………………………

Reviewer #3 comments

Reviewer #3: I only have minor comments for this version.

Lines 477-490: when it comes to the discussion of percent resistant isolates, I think the authors should be much more cautious in their statements. What is the meaning of 100% resistant isolates for a particular antibiotic when only one or two strains of a species have been isolated? Not much. If read as such, one could take as a general message that all Serratia spp. or Citrobacter freundii isolated from milk are resistant to tetracycline. Drawing such a conclusion from 1 or 2 isolates is really misleading. I think the authors should somehow nuance their remarks and focus on species for which they have at least 10 isolates.

Response: We have revised this. Serratia spp. and Citrobacter freundii with 4 and 2 isolates respectively have been excluded in the discussion. See L477-488

………………………………………………………………………………………………………

Reviewer #3: Line 475: the statement "Thus, it is important to isolate the coliform mastitis pathogen and perform an antibiotic susceptibility test before any antibiotic therapy." is not always realistic when it comes to application on the field. Very often, when a clinical mastitis case is detected, antibiotics are used without any prior identification of the pathogen. I am not saying it is not important to follow antibiotic resistance of mastitis pathogens, I'm rather implying that it not easily feasible in the field and that it is more a question for surveillance authorities. This statement should be somehow modified.

Response: Thank you, this statement has been revised. See L475-476 

………………………………………………………………………………………………………

---

## [Editor Report · Decision Letter 2]

26 Apr 2022

Prevalence and Risk Factors of Coliform-Associated Mastitis and Antibiotic Resistance of Coliforms from Lactating Dairy Cows in North West Cameroon

PONE-D-21-33571R2

Dear Dr. Ndip,

We’re pleased to inform you that your manuscript has been judged scientifically suitable for publication and will be formally accepted for publication once it meets all outstanding technical requirements.

Kind regards,

Pierre Germon

Academic Editor

PLOS ONE
---

## [Editor Report · Acceptance letter]

15 Jul 2022

PONE-D-21-33571R2 

Prevalence and Risk Factors of Coliform-Associated Mastitis and Antibiotic Resistance of Coliforms from Lactating Dairy Cows in North West Cameroon 

Dear Dr. Ndip:

I'm pleased to inform you that your manuscript has been deemed suitable for publication in PLOS ONE. Congratulations! Your manuscript is now with our production department. 

Kind regards, 

on behalf of

Dr. Pierre Germon 

Academic Editor

PLOS ONE